# Optimize Weight Rounding via Signed Gradient Descent for the Quantization of LLMs

## Abstract

Large Language Models (LLMs) have proven their exceptional capabilities in performing language-related tasks. However, their deployment poses significant challenges due to their considerable memory and storage requirements. In response to this issue, weight-only quantization, particularly 3 and 4-bit weight-only quantization, has emerged as one of the most viable solutions. As the number of bits decreases, the quantization grid broadens, thus emphasizing the importance of up and down rounding. While previous studies have demonstrated that fine-tuning up and down rounding with the addition of perturbations can enhance accuracy in some scenarios, our study is driven by the precise and limited boundary of these perturbations, where only the threshold for altering the rounding value is of significance. Consequently, we propose a concise and highly effective approach for optimizing the weight rounding task. Our method, named SignRound, involves lightweight block-wise tuning using signed gradient descent, enabling us to achieve outstanding results within 400 steps. SignRound competes impressively against recent methods without introducing additional inference overhead. The code will be open-sourced.

## 1 Introduction

Large language models (LLMs) have demonstrated exceptional proficiency on language-related tasks(OpenAI; Touvron et al., 2023a). Nevertheless, the deployment of LLMs presents notable hurdles due to their extensive memory and storage needs. Moreover, the computational demands of these models leads to the challenges for real-time applications. Consequently, it becomes imperative to explore techniques like quantization to facilitate the efficient deployment of LLMs.

Quantization techniques can be broadly classified into two categories: quantization-aware training (QAT) (Esser et al., 2019; Zhuang et al., 2021; Lee et al., 2021; Liu et al., 2023) and post-training quantization (PTQ) (Nagel et al., 2019; Xiao et al., 2022; Frantar et al., 2022; Nagel et al., 2020). QAT involves training the model with quantization in mind. During QAT, the model is trained using simulated lower-precision representations, allowing it to learn and adapt to the effects of quantization. This approach often yields better accuracy compared to PTQ. However, QAT comes with certain drawbacks, including increased training complexity, longer training times, and the need to tune hyperparameters. Applying QAT to LLMs can be particularly costly, despite recent efforts (Hu et al., 2021; Dettmers et al., 2023) to improve the efficiency of fine-tuning LLMs. In contrast, PTQ directly quantizes the model without any simulated training or fine-tuning. While PTQ is a concise approach, it is susceptible to significant accuracy drops. This highlights the need for further advancements in PTQ methods to enhance their accuracy preservation capabilities.

Two types of tensors could be quantized: activations and weights. Weight-only quantization has gained prominence in recent times as it offers a favorable tradeoff for LLMs. Quantizing activations for LLMs can be challenging (Wei et al., 2022b; Xiao et al., 2023; Bondarenko et al., 2023), making weight-only quantization a more practical choice. Additionally, the primary bottleneck in generating new tokens for LLMs often lies in memory bandwidth (Kim et al., 2023), further emphasizing the significance of weight-only quantization. In this work, we only focus on weight only quantization.

In order to quantize the weights, a rounding operation is necessary, with rounding-to-nearest (RTN) being the predominant method. RTN quantizes each element independently by simply rounding it to

the nearest integer. However, RTN fails to consider the relationships between weights and weights, as well as weights and activations. The potential of an advanced rounding strategy to improve accuracy has been initially demonstrated by Nagel et al. (Nagel et al., 2020). They addressed the rounding task by formulating it as a quadratic unconstrained binary optimization problem and approximated the task loss by employing a Taylor series expansion. However, relying exclusively on the second-order term may not produce accurate results. This is because rounding can introduce considerable weight modifications that may make other order terms significant and non-negligible.

We prefer the signed gradient descent method to effectively tackle the issue of sub-optimal rounding solutions. This approach is inspired by the well-defined boundaries of the solution space, which are confined to the range of [-0.5, 0.5], where only the threshold for altering the rounding value is of significance. Firstly, the optimal value is not a single float but typically a large region, negating the need for the gradient magnitude to converge to an exact point. Secondly, due to the limited boundary, we can traverse this space within a constrained number of steps, while the gradient magnitude may vary significantly, making it challenging to determine an appropriate step size within a limited number of iterations. Thirdly, signed SGD is inherently intuitive, allowing for easy adjustment of the step size (learning rate). For instance, we employed the same optimizer hyperparameters in all of our experiments, which include 400 steps and a learning rate of 0.0025, with linear weight decay. This ensures that 400*0.0025/2=0.5 covers the range of [-0.5, 0.5]. Figure 1 provides an overview of our method, SignRound. It utilizes signed gradient descent to fine-tune the up and down rounding through block-wise output reconstruction, resulting in enhanced flexibility and faster convergence. Our contributions are primarily threefold:

- We introduce a succinct and potent method for optimizing the weight-rounding task. Our approach utilizes a minimal amount of unlabeled data and executes quantization in a block-wise fashion. Moreover, it is worth noting that our method does not introduce any additional overhead during inference, further enhancing its general practicality.

- Our findings demonstrate that a mere alteration of approximately 5% of the rounding values can significantly enhance the performance of some quantization models.

- Our empirical results exhibit substantial performance enhancements over the established baseline of RTN, and our method contends favorably against recent techniques.

## 2 RELATED WORK

**Quantization Aware Training.** QAT methods have gained widespread popularity in model compression, as they enable the fine-tuning process, often leading to superior accuracy compared to the PTQ method. In their work, (Esser et al., 2019) proposed a novel approach that estimates and scales the task loss gradient at each weight and activation layer's quantizer step size, allowing for joint learning with other network parameters. (Zhuang et al., 2021) put forward a progressive quantization scheme that involves quantizing activations after weights. Additionally, CPQ (Lee et al., 2021) effectively identified the optimal quantization grids while naturally encouraging the underlying full-precision weights to gather around those quantization grids cohesively during training. While QAT methods are popular in relatively small-scale models, their application in LLMs is limited due to the high computational cost associated with training or fine-tuning.

**Post-training Quantization (PTQ).** PTQ methods simplify the quantization process without the needs of additional training. (Nagel et al., 2019) focused on minimizing quantization error through weight equalization and bias correction techniques. (Liu et al., 2021) specifically addressed the quantization of vision transformers, introducing a ranking loss to preserve the relative order of self-attention results after quantization and exploring a mixed-precision quantization scheme. (Frantar & Alistarh, 2022) leveraged Optimal Brain Surgeon (Hassibi et al., 1993) to tune weights during model compression. Both Hawq (Yao et al., 2021) and HAQ (Wang et al., 2019) aimed to identify important layers and maintain higher precision for them. Given its low resource requirement, PTQ is particularly suitable for the quantization of Large Language Models (LLMs). We will next focus on the quantization methods designed for LLMs, most of which fall under the category of PTQ.

**Large Language Models Quantization.** Significant advancements have been made in addressing the pressing demand for quantizing large language models (LLMs). LLM.int8() (Dettmers et al.,

2022) introduced a mixed precision approach to preserve essential channels in high precision. Zero-QuantV2 (Yao et al., 2023) employed low-rank matrices to enhance model quality recovery. RPTQ (Yuan et al., 2023) mitigated the impact of range differences between channel by rearranging the channels and quantizing them in clusters. Other methods, such as SPIQ (Yvinec et al., 2023), SmoothQuant (Xiao et al., 2022), Outlier Suppression+ (Wei et al., 2023), utilized handcrafted equivalent transformations to mitigate quantization errors. While these approaches are effective, their applicability is limited due to the performance overhead involved during inference, because there is no chance to fuse the transformation scale to the model itself on certain model architectures. LLM-QAT (Liu et al., 2023) employs QAT to enhance the performance of W4A8. In the context of weight-only quantization, GPTQ (Frantar et al., 2022) optimized weights using the Optimal Brain Surgeon (Hassibi et al., 1993) technique, achieving low-bit quantization on LLMs with minimal computational overhead. AWQ (Lin et al., 2023) followed the equivalent transformation approach with additional tuning in a constrained space, and has the similar limitations as SmoothQuant (Xiao et al., 2022). SqueezeLLM (Kim et al., 2023) employed sensitivity-based non-uniform quantization and dense-and-sparse decomposition to achieve lossless compression to ultra-low precision. While recent advancements in LLM quantization have made significant progress, there is still room for improvement in achieving minimal quantization loss without introducing inference overhead.

**Rounding Methods.**   Adaptive Rounding (Nagel et al., 2020) has already showcased the potential of an advanced rounding strategy to enhance accuracy (Li et al., 2021; Wei et al., 2022a). They used the rounding task as a quadratic unconstrained binary optimization problem by approximating the task loss through a Taylor series expansion. However, considering only the second-order term may not yield accurate results. This is because the rounding value gets multiplied by a scaling co-efficient during de-quantization, potentially introducing significant weight changes that make other order terms non-negligible. FlexRound (Lee et al., 2023) introduces a more flexible approach to rounding by incorporating element-wise division. This allows for simultaneous learning of a shared quantization grid size and individual scales for each pre-trained weight. However, it's not easily scalable to apply to LLMs due to the needs of specialized hyperparameters for each specific model and task. AQuant (Li et al., 2022) introduced a dynamic approach where the border becomes a function dependent on the activation value to reduce the quantization error of activation. We specifically concentrate on the up and down rounding task for weight quantization in this work.

**Signed Gradient Descent.**   Signed gradient descent is not commonly utilized and is typically applied in specific scenarios, such as reducing communication costs. This is because signed gradient carries significantly less information compared to original gradient. Recent studies have shed light on the advantages of sign-based methods over gradient descent in certain conditions. Safaryan et al. (Safaryan & Richtárik, 2021) found that sign-based methods are preferable when the Hessian matrix is concentrated on its diagonal and the maximal eigenvalue is much larger than the average eigenvalue. Li et al. (Li et al., 2023) investigated a variant of sign-based gradient descent that exhibits faster convergence. Additionally, Safaryan et al. (Safaryan & Richtárik, 2021) proposed a stochastic sign descent with momentum, which converges under the standard bounded variance assumption with the optimal asymptotic rate. These findings contribute to a better understanding of the potential benefits and applications of signed gradient descent methods.

## 3   METHODOLOGY

We provide an overview of quantization before diving into the details of our approach. To quantize and de-quantize the weights, the following operation as shown in Eq.1 is used (disregarding zero point for simplicity).

$$\widetilde{W} = s * clip(\left\lfloor \frac{W}{s} \right\rceil, n, m), n, m \in \mathbb{N} \tag{1}$$

where $s$ is the quantization scale, which is a positive scalar value. However, it is important to mention that our method can be easily extended to cases where $s$ is a vector or tensor. And the rounding operation $\lfloor \cdot \rceil$ is typically performed using the RTN method. While RTN is a concise approach, it quantizes each element independently, thereby losing the ability to model the correlation among different weights or activations.

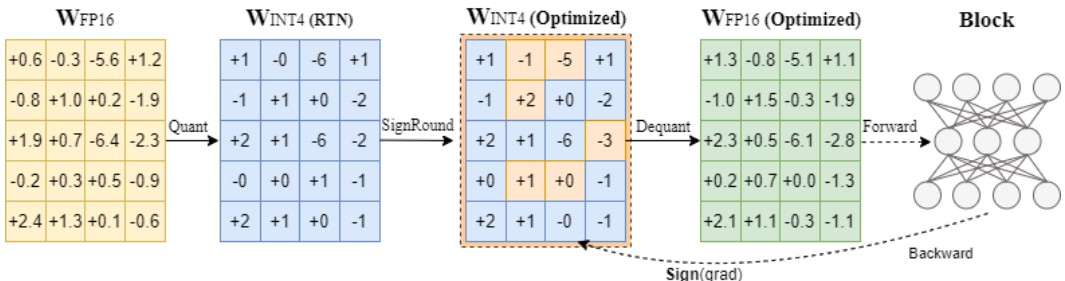

Figure 1: An illustration of SignRound. Unlike the direct rounding in RTN, SignRound performs signed gradient descent to fine-tune the up and down rounding through block-wise output reconstruction. After lightweight forward and backward steps, $\mathbf{W}_{\text{INT4}}$ has been well optimized towards the minimal loss, therefore ready for the final inference deployment. Note that Quant and Dequant are two standard operations for quantization and dequantization respectively.

To introduce more flexibility into the rounding operation, a tensor $V$ with the same shape of $W$ is introduced. Each element of $V$ falls within the range of $[-B, B]$, in which $B$ is set to 0.5 in all of experiments to ensure that the changes made only impact the rounding value.

$$\widetilde{W} = s * clip(\left\lfloor \frac{W}{s} + V \right\rceil, n, m), n, m \in \mathbb{N} \tag{2}$$

This adjustment allows for a more adaptable and context-aware quantization process. If we try to reconstruct the output of layers, the loss could be formulated as

$$L = ||WX - \widetilde{W}X||_F^2 \tag{3}$$

where $X$ is the input of the layer and $||\cdot||_F$ denotes the Frobenius norm. Then the final optimization task is described as the following

$$\arg\min_V ||WX - \widetilde{W}X||_F^2 \tag{4}$$

### 3.1 SIGNROUND

Since $V$ has a clear boundary, i.e. $[-0.5, 0.5]$, and only the threshold for altering the rounding value is of significance. These attributes provide three advantages for sign gradient descent as mentioned at the end of Section Introduction 1 . Therefore, we favor it as the optimizer. in Figure 1 shows an illustration of our method. More precisely, we follow the below optimization to approach the sub-optimal solution of Eq. 4.

$$V_{t+1} = V_t - lr * sign(\frac{\partial L}{\partial V})$$
$$\text{s.t.}|\sum_t lr * sign(\frac{\partial L}{\partial V})| \leq B \tag{5}$$

where $t$ is the optimizing step, lr is the learning rate, $|\cdot|$ is the absolute operation and B is the boundary we use, which is set to 0.5 in all our experiments.

Further, by employing straight-through estimator (STE) (Bengio et al., 2013), it can be easily demonstrated that $sign(\frac{\partial L}{\partial V}) = sign(\frac{\partial L}{\partial W})$ in Eq. 5 as following since elements of $s$ are all positive.

$$\frac{\partial L}{\partial W} = -2(WX - \widetilde{W}X)X^T \tag{6}$$

$$\frac{\partial L}{\partial V} = -2s(WX - \widetilde{W}X)X^T \tag{7}$$

---

**Algorithm 1** SignRound

---

**Input:** Calibration Data $\mathcal{D}$, learning rate $lr$, total steps $T$, Model $M$, block module $m_w$ with weights $w$, zero initialized $V$, batch size $bs$
**Output:** $best\_V$
1: $V \leftarrow 0, best\_V \leftarrow 0, best\_l \leftarrow maximum$
2: **for** $i \leftarrow 0$ to $T$ **do**
3:     $d \leftarrow draw\ bs\ samples\ from\ \mathcal{D}$
4:     $x \leftarrow M(d)_m$                                        ▷ get the inputs of m
5:     $y_f \leftarrow m_w(x)$                           ▷ get the output of original module
6:     $\widetilde{w} \leftarrow qdq(w, V)$                     ▷ quantize and dequantize w via Eq.2
7:     $y_q \leftarrow m_{\widetilde{w}}(x)$                     ▷ get the output of quantized module
8:     $loss \leftarrow mse(y_q, y_f)$                     ▷ get the loss via Eq.3
9:     **if** $loss < best\_l$ **then**
10:         $best\_V \leftarrow V$
11:         $best\_l \leftarrow loss$
12:     **end if**
13:     $loss.backward()$
14:     $update\ V\ via\ Eq.\ 8$
15: **end for**

---

So our optimization could be simplified as

$$V_{t+1} = V_t - lr * sign(\frac{\partial L}{\partial W})$$

$$\text{s.t.} |\sum_t lr * sign(\frac{\partial L}{\partial W})| \leq B \tag{8}$$

Moreover, as Eq 3 averages the loss of each element, which presumes that each one contributes equally to the network, that basically is not true. To alleviate this issue, we optimize the rounding task blockwise. To clarify, in our context, we use the term 'layer' to refer to a linear/convolution layer, while 'block' denotes a transformer block that typically consists of several linear layers.

The pseudocode 1 above provides additional details about SignRound. Inspired by recent research (Lin et al., 2023; Shao et al., 2023), we could also optimize weight min max tuning visa signed gradient descent, as described in A, which we refer to as **signroundv2** in our experiments.

## 4 EXPERIMENTS

In this section, we conduct a comprehensive evaluation of SignRound from various perspectives. Firstly, we provide a brief overview of the LLM architectures and tasks that are included in our evaluation. Secondly, we present a detailed comparison between our method and some other existing approaches, highlighting the unique features and advantages of SignRound. Thirdly, we conduct additional experiments to further demonstrate the validity of our choices, assess the sensitivity of hyperparameters, and explore other relevant factors. Finally, the runtime is reported in Appendix F for reference.

### 4.1 EXPERIMENTAL SETTINGS

**Evaluation and Datasets.** We make assessments on several language tasks to satisfy the task-agnostic setting. Specifically, we report average accuracy results on four common sense reasoning tasks including HellaSwag (Zellers et al., 2019), WinoGrande (Sakaguchi et al., 2021), PIQA (Bisk et al., 2020) and LAMBADA (Paperno et al., 2016).Additionally, we benchmarked our models on MMLU (Hendrycks et al., 2020), which encompasses 57 tasks spanning STEM, humanities, social science, and more. Lm-eval-harness (Gao et al., 2021) is adopted to perform the evaluation for all

these tasks. Furthermore, we complement our evaluation with perplexity (ppl) analysis on Wikitext2 (Merity et al., 2016) and C4 (Raffel et al., 2020), by following the source code [1] of GPTQ.

**Quantization Configurations.** In line with the approach taken in GPTQ (Frantar et al., 2022), we specifically concentrate on weight-only quantization, targeting the linear layers within transformer blocks. Other layers, such as the embedding layer and typically the last layer like lm-head, are excluded from the quantization process. We initially intended to utilize the pile (Gao et al., 2020) dataset for calibration, following AWQ (Lin et al., 2023) and SmoothQuant (Xiao et al., 2022). However, due to its large size, we have opted to use the readily available pile-10k dataset [2], which consists of the first 10k samples from pile, for both GPTQ and our method. We employ standard uniform per-row asymmetric quantization on the min-max grid. Our evaluation primarily focuses on W4, W4G128, and W3G128, where W4 indicates quantizing weights with 4 bits and G represents finer-granularity grouping as described in (Park et al., 2022; Frantar et al., 2022).

**Large Language Models.** Our experimental evaluation encompasses a range of widely adopted LLM architectures, such as LLaMAs (Touvron et al., 2023a), LLaMAs v2 (Touvron et al., 2023b), BLOOMs (Scao et al., 2022), and OPTs (Zhang et al., 2022). We cover a wide range of LLM parameters, ranging from millions to billions, to ensure comprehensive coverage and analysis.

**SignRound Hyperparameters.** We selected 512 samples randomly from pile-10k and truncated each sample to a sequence length of 512. The tuning process involves adjusting each block for 400 steps using a learning rate of 2.5e-3, a batch size of 8, and employing a linear learning rate decay. We set the value of B in Eq. 8 to 0.5. Besides, we adopted automatic mixed precision(AMP) to accelerate the tuning. It's worth noting that adjusting the sequence length to 2048 yielded improvements in numerous scenarios. However, we did not adopt this as the default setting due to the associated runtime overhead. For models $\geq$ 30B, we made configuration adjustments to strike a balance between runtime and performance. Specifically, we reduced the sample count to 256, shorted the sequence length to 256, and disabled AMP.

Table 1: Average % accuracy(↑) of HellaSwag, WinoGrand, PIQA and LAMBADA for LLaMA & OPT. "Oursv2" denotes incorporating minmax tuning detailed in Appendix A.As demonstrated in Table 13, For models $\leq$ 13B, the runtime of ours matches that of GPTQ without actorder, while oursv2 aligns with GPTQ-R that with actorder.

| nbits | methods | LLaMA | | | | OPT | | | | |
|---|---|---|---|---|---|---|---|---|---|---|
| | | 7b | 13b | 7bv2 | 13bv2 | 125m | 1.3b | 2.7b | 6.7b | 13b |
| 16 | FP16 | 68.80 | 71.14 | 69.02 | 71.20 | 45.09 | 57.66 | 61.04 | 64.92 | 65.49 |
| W4 | RTN | 67.38 | 68.82 | 66.98 | 70.17 | 39.41 | 47.22 | 58.61 | 62.99 | 64.08 |
| | GPTQ | 64.70 | 70.00 | 66.89 | 69.24 | 43.58 | 56.15 | 59.92 | 63.09 | 64.83 |
| | GPTQ-R | 67.71 | 70.06 | 67.62 | 70.25 | 43.97 | 56.11 | **60.66** | 63.85 | 64.93 |
| | Ours | 68.05 | 70.58 | 67.74 | 70.03 | 44.13 | 56.17 | 60.58 | 64.34 | 65.05 |
| | Ours-v2 | **68.31** | **70.87** | **68.16** | **71.18** | **45.26** | **57.20** | 60.20 | **64.70** | **65.37** |
| W4G128 | RTN | 67.85 | 70.84 | 68.32 | 70.72 | **45.27** | 56.47 | 60.70 | 64.03 | 64.84 |
| | GPTQ | 66.32 | 70.92 | **68.90** | 70.68 | 42.88 | 56.99 | **61.23** | 64.75 | 65.37 |
| | GPTQ-R | 68.48 | 71.07 | 68.69 | 70.59 | 43.01 | 57.47 | 60.68 | 64.75 | 65.62 |
| | Ours | 68.09 | **71.43** | 68.65 | 70.81 | 44.23 | 57.30 | 60.86 | 64.76 | **65.67** |
| | Ours-V2 | **68.68** | 71.12 | 68.60 | **71.12** | 44.70 | **57.76** | 60.87 | **65.12** | 65.42 |
| W3G128 | RTN | 64.94 | 67.70 | 65.92 | 68.70 | 39.11 | 42.61 | 36.99 | 56.09 | 49.56 |
| | GPTQ | 58.29 | 68.73 | 65.51 | 68.73 | 39.78 | 54.43 | 58.47 | 62.98 | **64.68** |
| | GPTQ-R | 66.46 | 69.14 | 66.50 | 69.53 | 42.56 | 54.91 | 59.78 | 63.00 | 64.47 |
| | Ours | 66.62 | 69.59 | 66.88 | 69.70 | 43.31 | 55.46 | 59.12 | 53.42 | 63.61 |
| | Ours-v2 | **66.78** | **70.13** | **67.25** | **70.45** | **44.19** | **55.78** | **60.33** | **64.62** | 64.46 |

---

[1] https://github.com/IST-DASLab/gptq

[2] https://huggingface.co/datasets/NeelNanda/pile-10k

Table 2: Average % accuracy(↑) of HellaSwag, WinoGrand, PIQA and LAMBADA for BLOOM. There is no act-order option in GPTQ for Bloom models.

| | W4 | | | | W4G128 | | | | W3G128 | | | |
|---|---|---|---|---|---|---|---|---|---|---|---|---|
| Size | 560m | 1b7 | 3b | 7b1 | 560m | 1b7 | 3b | 7b1 | 560m | 1b7 | 3b | 7b1 |
| FP16 | 45.50 | 52.31 | 55.48 | 60.22 | 45.50 | 52.31 | 55.48 | 60.22 | 45.50 | 52.31 | 55.48 | 60.22 |
| RTN | 43.10 | 49.97 | 53.16 | 57.73 | 44.28 | **52.08** | 54.86 | 59.31 | 40.83 | 47.98 | 52.51 | 57.59 |
| GPTQ | 43.95 | 50.91 | 54.65 | 58.27 | 44.79 | **52.08** | 55.68 | 59.59 | 42.74 | 48.81 | 53.41 | 58.12 |
| Ours | **45.00** | 51.47 | 54.63 | 59.52 | **45.40** | 51.85 | 55.40 | 59.83 | 44.08 | 50.52 | 53.64 | 58.69 |
| Ours-v2 | 44.97 | **51.85** | **55.33** | **59.64** | 45.31 | 51.66 | **55.73** | **59.92** | **44.27** | **51.38** | **54.30** | **59.14** |

Table 3: C4 ppl ( ↓) at W4. There is no act-order option in GPTQ for Bloom models.

| | LLaMA | | | | OPT | | | | BLOOM | | | |
|---|---|---|---|---|---|---|---|---|---|---|---|---|
| Size | 7b | 13b | 7bv2 | 13bv2 | 1.3b | 2.7b | 6.7b | 13b | 560m | 1b7 | 3b | 7b1 |
| FP16 | 7.34 | 6.80 | 7.26 | 6.73 | 16.07 | 14.34 | 12.71 | 12.06 | 26.59 | 19.49 | 17.48 | 15.20 |
| RTN | 8.12 | 7.23 | 8.16 | 7.14 | 27.49 | 18.83 | 14.37 | 13.32 | 29.87 | 21.25 | 18.76 | 16.05 |
| GPTQ | 8.64 | 7.13 | **7.90** | **6.87** | 17.04 | 15.06 | 13.39 | 12.29 | 28.15 | 20.71 | 18.18 | 15.67 |
| GPTQ-R | 7.82 | 7.10 | 7.83 | 7.06 | 16.84 | 14.95 | 13.05 | **12.27** | - | - | **-** | **-** |
| Ours | 7.84 | 7.05 | 11.20 | 7.72 | 16.92 | 14.97 | 13.08 | 12.48 | 28.12 | 20.41 | 18.18 | 15.67 |
| Ours-v2 | **7.67** | **6.95** | 9.89 | 6.91 | **16.63** | **14.70** | **12.92** | 12.32 | **27.69** | **20.14** | **17.95** | **15.53** |

## 4.2 COMPARING WITH OTHER METHODS

We conducted a comprehensive benchmarking of our results against RTN and GPTQ (Frantar et al., 2022). We denote GPTQ with actorder as GPTQ-R. Also, it's worth noting that bloom models do not offer this option in GPTQ. When evaluating perplexity (ppl), we prioritize reporting the ppl on C4 dataset as our primary focus, taking into consideration the potential occurrence of NaN values when assessing perplexity for Wikitext2 and ptb datasets, both for SignRound and GPTQ. Furthermore, we conducted a limited and non-rigorous comparison between our approach and AWQ Lin et al. (2023) in Appendix B.1.

We begin by presenting the average accuracy results for the HellaSwag, WinoGrand, PIQA, and LAMBADA tasks across LLaMA, OPT, and BLOOM models with a size below 13B. These results are shown in Table 1 and 2.In summary, SignRound demonstrates superior performance over RTN in 36 out of 39 scenarios, highlighting its effectiveness. Furthermore, when compared to best(GPTQ,GPTQ-R), our approach surpasses it in 27 out of 39 scenarios. Additionally, when incorporating signroundv2, we outperform it in 34 out of 39 scenarios, further emphasizing the strengths of our method. For detailed results of LLaMA7B, LLaMA13B, LLAMA7B-V2, and LLAMA13B-V2, please refer to Appendix G. The results in Appendix G also highlight that changing the sequence length to 2048 could bring noticeable improvement in many scenarios.

We then present the perplexity (ppl) results for C4 in Table 3, along with the detailed results for Wikitext2 in Appendix B.2. In conclusion, we achieve better performance in 9 out of 12 models. However, there are some outliers. With the observation that the c4 perplexity of GPTQ for LLaMAv2 is reasonable, while its ptb perplexity is NAN in Table 16, and Signround has the similar issue, we hypothesize this is likely because that perplexity is highly sensitive to outliers. The equation is represented similar to $\exp(-(sum(\log p))/seqlen)$, where a low probability p for a single token results in a high perplexity value. In certain cases where the results may not be optimal, we can still fine-tune the hyperparameters to achieve better results, as demonstrated in the subsequent sections.

Next, we present a comprehensive breakdown of the accuracies achieved by MMLU for LLaMA-7B and LLaMa-7B-V2 in Table 4. By analyzing the average accuracies, we observe that SingRound outperforms RTN and GPTQ in 4 out of the 6 scenarios when the best model-wise setting is applied.

We also provide the results for models with a capacity of 30B or greater at W3G128 in Table 5 and W4 in Appendix B.3. Additionally, we discovered that recovering the sequence length to 512 of the calibration dataset yielded improvements in certain scenarios, and thus we include these results. In summary, our approach achieves comparable performance to GPTQ for the given accuracy task. However, we slightly lag behind GPTQ in terms of ppl tasks.

### 4.3 ABLATION STUDY OF OPTIMIZERS

We investigated the impact of various optimizers in Table 6 . Given the sensitivity of perplexity as mentioned earlier, our primary focus was on average accuracies. We found that signed gradient descent outperformed SGD and was comparable to ADAMW (Loshchilov & Hutter, 2017). Notably, ADAMW was shown to be 20%-30% slower, as indicated in Table 13, and required significantly more memory, which is crucial for large language models. Consequently, signed gradient descent offers distinct advantages in this context. For some other ablation studies, please refer to Appendix C.

### 4.4 THE ANALYSIS OF GRADIENTS AND THEIR EFFECTS ON ROUNDING

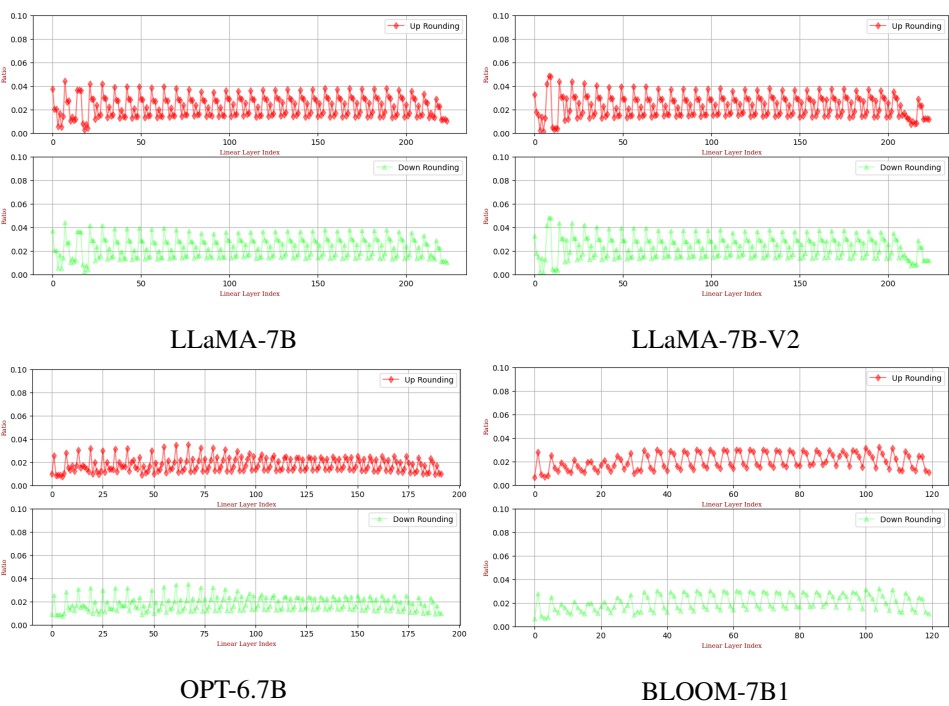

Figure 2: The impact of the rounding value introduced by the $V$ in Eq. 2

Table 4: Accuracies(↑) of MMLU(5-shot) for LLaMA-7B & LLaMA-7B-V2. "Ours-2048" indicates that we have modified the sequence length of the calibration dataset from 512 to 2048.

| | | LLaMA-7B | | | | | LLaMA-7B-V2 | | | | |
| | | Hums. | STEM | Social | Other | Avg. | Hums. | STEM | Social | Other | Avg. |
|---|---|---|---|---|---|---|---|---|---|---|---|
| | FP16 | 38.32 | 31.17 | 38.05 | 36.85 | 35.64 | 51.40 | 37.00 | 52.23 | 49.51 | 46.56 |
| W4G-1 | RTN | 34.84 | 29.53 | 32.87 | 36.28 | 33.10 | 44.03 | 32.83 | 44.97 | 42.19 | 40.24 |
| | GPTQ | 33.31 | 26.29 | 29.86 | 33.11 | 30.32 | 46.21 | **34.29** | 46.68 | 44.85 | 42.21 |
| | Ours | 34.30 | **31.05** | 34.74 | 36.66 | 33.95 | 47.28 | 33.14 | 46.90 | 44.70 | 42.10 |
| | Ours2048 | **35.10** | 30.69 | **36.43** | **36.85** | **34.42** | **47.40** | 33.92 | **49.61** | **44.91** | **43.00** |
| W4G128 | RTN | 36.30 | **31.67** | **37.40** | **37.99** | **35.48** | 49.54 | 36.50 | 50.95 | 47.87 | 45.31 |
| | GPTQ | **37.77** | 29.64 | 36.38 | 37.45 | 34.83 | 50.30 | 36.51 | 50.91 | 47.69 | 45.43 |
| | Ours | 36.06 | 30.86 | 35.99 | 36.21 | 34.44 | **51.39** | **37.87** | **52.56** | **49.69** | **46.95** |
| | Ours2048 | 35.66 | 30.05 | 36.16 | 37.57 | 34.46 | 50.12 | 36.70 | 51.44 | 48.20 | 45.69 |
| W3G128 | RTN | **32.97** | **30.28** | **33.66** | **32.60** | **32.17** | 41.14 | 33.06 | 40.98 | 40.94 | 38.51 |
| | GPTQ | 30.77 | 28.29 | 30.73 | 31.33 | 30.12 | **44.66** | **37.55** | **46.36** | 43.47 | **42.48** |
| | Ours | 30.12 | 28.21 | 30.64 | 30.34 | 29.68 | 44.53 | 33.53 | 44.60 | **43.52** | 40.82 |
| | Ours2048 | 32.43 | 28.62 | 31.03 | 32.10 | 30.85 | 42.75 | 32.98 | 42.88 | 41.30 | 39.34 |

Table 5: Average % accuracy(↑) of HellaSwag, WinoGrand, PIQA and LAMBADA and C4 ppl(↓) for LLaMA & OPT with ≥ 30B at W3G128. "Ours-seq512" indicates that we have modified the sequence length of the calibration dataset from 256 to 512.

| | Accuracy | | | | PPL on C4 | | | |
|---|---|---|---|---|---|---|---|---|
| | LLaMA | | OPT | | LLaMA | | OPT | |
| Type | 30b | 65b | 30b | 66b | 30b | 65b | 30b | 66b |
| Size | | | | | | | | |
| FP16 | 73.46 | 75.48 | 67.87 | 69.54 | 6.13 | 5.98 | 11.46 | 10.99 |
| RTN | 72.17 | 73.69 | 62.83 | 38.00 | 6.85 | 6.52 | 30.81 | 285.41 |
| GPTQ | 72.09 | **73.97** | 66.76 | 67.87 | 6.80 | 6.52 | **11.74** | 11.87 |
| GPTQ-R | 71.46 | 73.68 | **67.12** | **68.98** | 6.77 | 6.43 | **11.73** | **11.20** |
| Ours-seq256 | **72.45** | 73.71 | 66.51 | 68.00 | 6.83 | **6.52** | 13.00 | 13.34 |
| Ours-seq512 | 71.95 | 73.78 | 66.70 | 67.26 | **6.79** | 6.53 | 12.50 | 13.97 |
| Oursv2-seq512 | 72.41 | 73.77 | **67.30** | **69.19** | **6.58** | **6.47** | 11.83 | 11.47 |

Table 6: Comparing different optimizers for around 7B models at W4G-1, the models LLaMA7b, LLaMA7bv2, OPT6.7b, and BLOOM7b1 are denoted by 7b, 7bv2, 6.7b, and 7b1 respectively. The accuracy is the % average accuracy(↑) of HellaSwag, WinoGrand, PIQA and LAMBADA . Perplexity (PPL) (↓) is evaluated using the C4 dataset. "wd" denotes weight decay. All other hyperparameters remain the same.

| | lm-eval | | | | C4 PPL | | | |
|---|---|---|---|---|---|---|---|---|
| Size | 7b | 7bv2 | 6.7b | 7b1 | 7b | 7bv2 | 6.7b | 7b1 |
| sgd-wd-0 | 66.73 | 67.23 | 63.10 | 57.88 | 8.09 | 8.19 | 14.07 | 16.02 |
| adamw-wd-1e-2 | 67.99 | 67.66 | **64.78** | 59.36 | 7.83 | 11.27 | **13.03** | **15.67** |
| adamw-wd-0 | 67.61 | **67.95** | 64.64 | 59.05 | **7.80** | **10.50** | **13.03** | **15.67** |
| ours | **68.05** | 67.74 | 64.34 | **59.52** | 7.84 | 11.20 | 13.08 | **15.67** |

In this analysis, we dive into the distribution of the magnitude of $V$ in Eq. 2 and its impact on rounding values across approximately 7 billion models at W4. The visual representations of these distributions are provided in Appendix D. Our investigation reveals that the majority of V values are concentrated within the range of [-0.3, 0.3]. Additionally, we observe an interesting pattern in the distribution of V across different layers. The middle layers exhibit a more tightly clustered distribution compared to the other layers. This observation aligns with the common understanding that the head and tail layers tend to be more sensitive to compression, while the middle layers are relatively more robust.

Figure 2 illustrates the impact of the rounding value introduced by the $V$ in Eq. 2 for models around 7B at W4. The red line represents "up rounding", indicating that while RTN rounds the value to the floor, SignRound changes it to the ceiling. Conversely, the green line represents "down rounding" indicating that while RTN rounds the value to the ceiling, SignRound changes it to the floor. It is worth noting that SignRound modifies only a small percentage of weight rounding values for each of the four models, namely 5.27%, 5.29%, 4.14%, and 4.10%.

We were also intrigued by the possible correlation between rounding and activation, we shown the result in Appendix E.

## 5 CONCLUSIONS AND LIMITATIONS

In this paper, we present a highly effective and concise approach to optimize the weight rounding task. Our method, SignRound, leverages lightweight block-wise tuning using signed gradient descent, achieving remarkable results within a mere 400 steps. Extensive experiments demonstrate the superior performance of our approach. As part of our future work, we plan to contribute our recipes and implementations to the open source community. On the other hand, although our method is generally effective, there are a few outliers in certain scenarios, where we plan to mitigate the issue by fine-tuning the hyperparameters.

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

## A  COMBINING WITH WEIGHT MINMAX TUNING

Drawing inspiration from recent works such as (Lin et al., 2023) and (Shao et al., 2023), and given the simplicity of our method, we can seamlessly incorporate weight minmax tuning using sign gra-dient descent. This integration allows us to observe significant improvements without introducing significant runtime overhead. we extend E.q. 1 to the following

$$s = \frac{max(W) - min(W)}{2^{bit} - 1} \tag{9}$$

$$\widetilde{W} = s * clip(\left\lfloor \frac{W}{s} + zp \right\rceil, n, m), n, m \in \mathbb{N} \tag{10}$$

in which zp is the zero point.

To enhance the effectiveness of the rounding quantization operation, we introduce two more trainable parameters, namely $\alpha, \beta$ which are incorporated into the above equations like the following

$$s = \frac{max(W) * \alpha - min(W) * \beta}{2^{bit} - 1} \tag{11}$$

and

$$\widetilde{W} = s * clip(\left\lfloor \frac{W}{s} + zp + V \right\rceil, n, m), n, m \in \mathbb{N} \tag{12}$$

We enforce the values of $\alpha$ and $\beta$ to be within the range $(0, 1]$ and use the same learning rate of 0.0025. For large models $\geq$ 30b, we kept the same setting as small models, i.e., sample count 512, sequence length 512 unless explicitly stated and enable AMP. All other settings remain unchanged.

## B  MORE RESULTS

### B.1  COMPARISON WITH AWQ

We present the results in comparison to AWQ in Table 7, all of which have been tested using the same calibration dataset pile-10k.

Table 7: Average % accuracy(↑) of HellaSwag, WinoGrand, PIQA and LAMBADA for LLaMA with AWQ and Ours. "Ours-v2" denotes incorporating minmax tuning detailed in Appendix A

| LLaMA | | 7b | 13b | 7bv2 | 13bv2 |
|---|---|---|---|---|---|
| FP16 | | 68.8 | 71.14 | 69.02 | 71.2 |
| W4 | AWQ | 67.51 | 70.53 | 68.06 | 70.39 |
| | Ours | 68.05 | 70.58 | 67.74 | 70.03 |
| | Ours-v2 | **68.31** | **70.87** | **68.16** | **71.18** |
| W4G128 | AWQ | 68.21 | 71.23 | 68.54 | 70.92 |
| | Ours | 68.09 | **71.43** | **68.65** | 70.81 |
| | Ours-v2 | **68.68** | 71.12 | 68.60 | **71.12** |
| W3G128 | AWQ | 66.35 | **70.37** | 66.52 | 69.63 |
| | Ours | 66.62 | 69.59 | 66.88 | 69.70 |
| | Ours-v2 | **66.78** | 70.13 | **67.25** | **70.45** |

### B.2  RESULTS OF WIKITEXT2 PPL AT W4

The perplexity results for Wikitext2 at W4 are shown in Table 8. In conclusion, our performance is comparable to that of GPTQ. "Oursv2" denotes incorporating minmax tuning detailed in Appendix A

Table 8: Wikitext2 ppl (↓) at W4

| | LLaMA | | | | OPT | | | | BLOOM | | | |
|---|---|---|---|---|---|---|---|---|---|---|---|---|
| Size | 7b | 13b | 7bv2 | 13bv2 | 1.3b | 2.7b | 6.7b | 13b | 560m | 1b7 | 3b | 7b1 |
| FP16 | 5.67 | 5.09 | 5.47 | 4.88 | 14.62 | 12.47 | 10.86 | 10.13 | 22.41 | 15.39 | 13.48 | 11.37 |
| RTN | 6.29 | 5.53 | 6.12 | 5.20 | 48.20 | 16.92 | 12.10 | 11.32 | 25.88 | 16.97 | 14.75 | 12.10 |
| GPTQ | 6.59 | 5.33 | 6.09 | 5.16 | 15.67 | 13.30 | 11.59 | **10.33** | 23.95 | 16.37 | 14.10 | 11.73 |
| GPTQ-R | 6.06 | 5.35 | **5.86** | **5.12** | 15.59 | 13.09 | 11.22 | 10.35 | - | - | - | - |
| Ours | 6.12 | 5.32 | 298 | 9.15 | 15.65 | 13.05 | 11.18 | 10.66 | 23.80 | 16.22 | 14.13 | 11.80 |
| Ours-v2 | **5.99** | **5.22** | 212 | 6.11 | **15.13** | **12.78** | **11.03** | 10.39 | **23.64** | **16.07** | **13.98** | **11.66** |

## B.3 Other results for large models

We present the results for models with a capacity of 30B or higher at W4 in Table 9 and PPL on Wikitext2 in Table 10. Furthermore, we observed that adjusting the sequence length of the calibration dataset led to improvements in specific scenarios, and we include these findings in our analysis. Overall, our approach demonstrates comparable accuracy performance to GPTQ for the given task. However, it is worth noting that we slightly fall behind GPTQ in terms of PPL tasks.

Table 9: Average % accuracy(↑) of HellaSwag, WinoGrand, PIQA and LAMBADA and C4 ppl(↓) for LLaMA & OPT with size ≥ 30B at W4. "Ours-seq512" indicates that we have modified the sequence length of the calibration dataset from 256 to 512. "Ours-v2" denotes incorporating minmax tuning detailed in Appendix A. Since Oursv2 is slower than GPTQ for modles ≥ 30B, we list their result separately.

| | Accuracy | | | | PPL on C4 | | | |
|---|---|---|---|---|---|---|---|---|
| Type | LLaMA | | OPT | | LLaMA | | OPT | |
| Size | 30b | 65b | 30b | 66b | 30b | 65b | 30b | 66b |
| FP16 | 73.46 | 75.48 | 67.87 | 69.54 | 6.13 | 5.98 | 11.45 | 10.99 |
| RTN | 72.33 | 73.91 | 65.94 | 37.12 | 6.54 | 6.46 | 13.56 | 305.73 |
| GPTQ | 72.85 | **74.45** | **67.55** | 68.23 | **6.42** | 6.23 | 11.59 | 11.24 |
| GPTQ-R | **72.94** | 74.42 | 67.41 | 68.84 | **6.42** | **6.19** | **11.58** | **11.09** |
| Ours-seq256 | 72.69 | 74.03 | 66.74 | 68.80 | 6.47 | 6.31 | 11.84 | 11.42 |
| Ours-seq512 | 72.86 | 73.91 | 67.40 | **69.22** | 6.47 | 6.34 | 11.77 | 11.45 |
| Oursv2-seq512 | 72.63 | 73.73 | 67.47 | 69.25 | **6.36** | 6.47 | 11.60 | 11.13 |
| Oursv2-seq2048 | **73.19** | 74.43 | **67.68** | 69.51 | **6.36** | 6.28 | **11.57** | 11.15 |

Table 10: Wikitext ppl(↓) for LLaMA & OPT with size ≥ 30B. "Ours-seq512" indicates that we have modified the sequence length of the calibration dataset from 256 to 512."Oursv2" denotes incorporating minmax tuning detailed in Appendix A

| | W4 | | | | W3G128 | | | |
|---|---|---|---|---|---|---|---|---|
| Type | LLaMA | | OPT | | LLaMA | | OPT | |
| Size | 30b | 65b | 30b | 66b | 30b | 65b | 30b | 66b |
| FP16 | 4.10 | 3.56 | 9.56 | 9.34 | 4.10 | 3.56 | 9.56 | 9.34 |
| RTN | 4.54 | 3.99 | 10.98 | 110.43 | 4.87 | 4.44 | 23.05 | 126.92 |
| GPTQ | 4.45 | 4.16 | **9.66** | 9.66 | 4.84 | 4.17 | **9.75** | 10.58 |
| GPTQ-R | **4.42** | 4.11 | 9.68 | **9.47** | 4.79 | 4.21 | 9.78 | **9.60** |
| Ours-seq256 | 4.51 | 3.91 | 9.88 | 9.56 | 4.85 | **4.15** | 11.07 | 11.40 |
| Ours-seq512 | 4.52 | **3.90** | 9.88 | 9.70 | **4.81** | 4.17 | 10.54 | 10.87 |
| Ours-v2-seq512 | **4.35** | **3.77** | 9.68 | **9.42** | **4.64** | **4.07** | 9.95 | 9.71 |

## C More Ablation studies

### C.1 Block-wise versus Layer-wise

We examined the effects of layer-wise and block-wise tuning. As explained in Section 3.1, the term "layer" refers to a linear/convolution layer, while "block" specifically denotes a transformer block consisting of multiple linear layers. To simplify this evaluation, we set the sequence length to 256 and disable AMP. Based on the below results, block-wise tuning outperformed layer-wise tuning in the majority of scenarios.

### C.2 The analysis of hyperparameters sensitivity

We conducted a hyperparameters sensitivity analysis, the results of which are summarized in Table 12. In the "steps100" configuration, we used 100 steps, and a learning rate of 1e-2. In the "lr4e-3"

Table 11: Comparing block-wise and layer-wise tuning for around 7B models, the models LLaMA7b, LLaMA7bv2, OPT6.7b, and BLOOM7b1 are denoted by 7b, 7bv2, 6.7b, and 7b1 respectively. The accuracy is the % average accuracy(↑) of HellaSwag, WinoGrand, PIQA and LAMBADA . Perplexity (PPL) (↓) is evaluated using the C4 dataset.

| | W4 | | | | W3G128 | | | |
|---|---|---|---|---|---|---|---|---|
| Size | 7b | 7bv2 | 6.7b | 7b1 | 7b | 7bv2 | 6.7b | 7b1 |
| layer-acc-seq256 | 67.50 | 67.78 | 63.46 | 58.72 | 65.96 | 66.09 | **61.60** | 58.24 |
| block-acc-seq256 | **67.64** | **67.96** | **64.55** | **59.08** | **66.31** | **66.63** | 57.76 | **58.34** |
| layer-c4-ppl-seq256 | 8.02 | **7.92** | 13.44 | 15.73 | 8.81 | **8.69** | **16.83** | 16.15 |
| block-c4-ppl-seq256 | **7.81** | 8.19 | **13.10** | **15.71** | **8.34** | 10.84 | 25.44 | **16.05** |

configuration, we set the learning rate to 4e-3. We also changed the sequence length of the calibration dataset from 512 to 2048, denoted by "seq2048". Please note that all other hyperparameters not mentioned in each configuration were kept the same as the default configurations, as detailed in Section 4.1. Overall, our method exhibits robustness to hyperparameters in common sense reasoning tasks, with the exception of the perplexity of LLaMA-7b-v2. However, we did discover that certain hyperparameters, such as the sequence length of the calibration dataset, can significantly impact performance in some scenarios, as demonstrated in Table 4 and 5.

Table 12: Hyperparameter sensitivity analysis, the models LLaMA7b, LLaMA7bv2, OPT6.7b, and BLOOM7b1 are denoted by 7b, 7bv2, 6.7b, and 7b1 respectively. The accuracy is the % average accuracy(↑) of HellaSwag, WinoGrand, PIQA and LAMBADA . Perplexity (PPL) (↓) is evaluated using the C4 dataset.

| | Accuracy | | | | PPL on C4 | | | |
|---|---|---|---|---|---|---|---|---|
| Size | 7b | 7bv2 | 6.7b | 7b1 | 7b | 7bv2 | 6.7b | 7b1 |
| steps100 | 67.53 | 67.76 | **64.64** | 58.76 | 7.93 | **7.83** | 13.12 | 15.71 |
| lr4e-3 | 68.01 | 67.57 | 64.57 | 59.47 | 7.81 | 10.29 | 13.09 | **15.66** |
| seq2048 | **68.11** | **67.79** | 64.32 | 59.39 | **7.76** | 9.97 | **13.06** | **15.66** |
| default | 68.05 | 67.74 | 64.34 | **59.52** | 7.84 | 11.20 | 13.08 | 15.67 |

## D    VISUALIZATION OF V

We provide an analysis of the magnitude distribution of V in Eq. 2 for approximately 7B models at W4 in Figure 3. The findings reveal that the majority of V values are concentrated within the range of [-0.3, 0.3]. Notably, the middle layers demonstrate a narrower distribution in comparison to the other layers. This observation suggests that the head or tail layers may be more susceptible to the compression.

## E    CORRECTION BETWEEN SIGNROUND AND SALIENT ACTIVATION CHANNELS

We were also intrigued by the possible correlation between rounding and activation, as previous research has shown that keeping only 0.1%-1% of the channels corresponding to larger activation can significantly improve the quantized performance in AWQ (Lin et al., 2023). Therefore, we investigated whether the altered rounding values tend to fall more frequently in these salient channels. The results of our analysis, presented in Figure 4, reveal an interesting finding. The ratio, representing the percentage of altered rounding values falling within the top 1% salient activation channels out of all altered rounding values, is typically around 1%. This suggests that there is no strong correlation between rounding and activation. It is possible that rounding values of less significant channels need to be changed to compensate for the quantization error introduced by these salient channels.

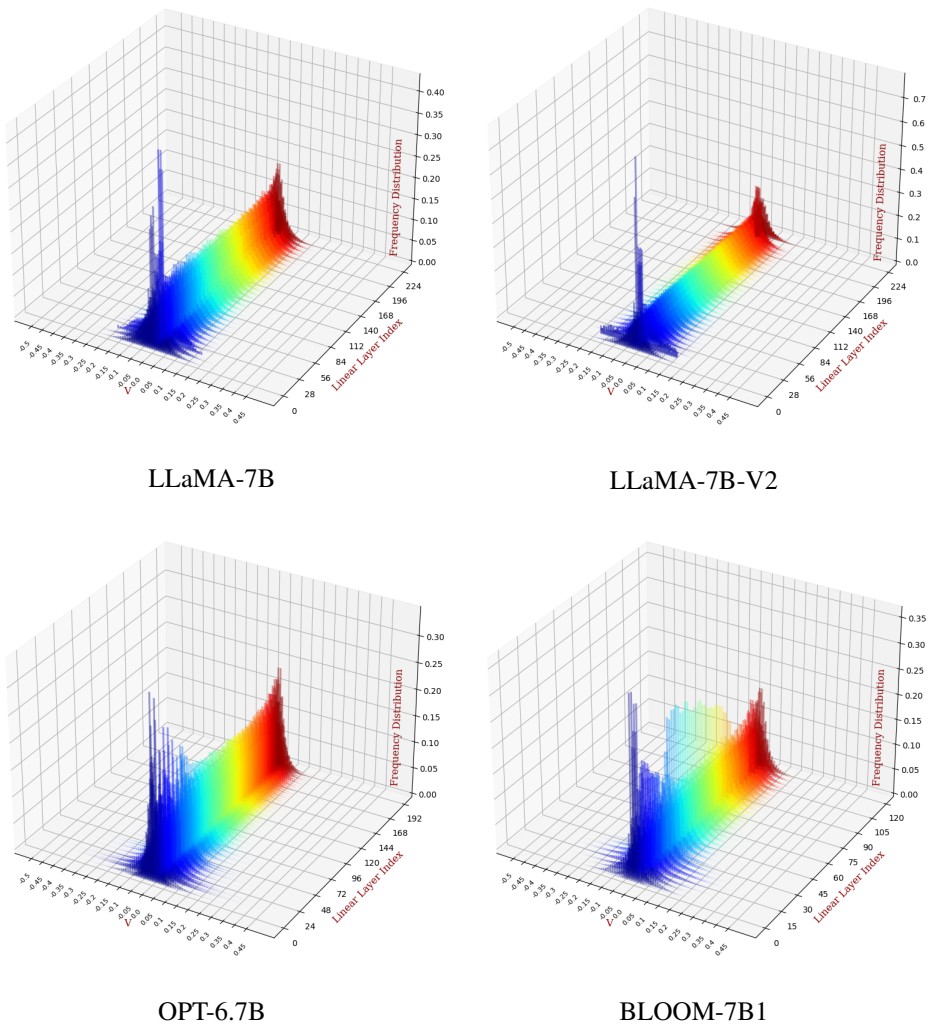

LLaMA-7B                    LLaMA-7B-V2

OPT-6.7B                    BLOOM-7B1

Figure 3: The distribution of the magnitude of V in Eq. 2 for different models, namely LLaMA-7B, LLaMA-7B-V2, OPT-6.7B, and BLOOM-7B1 at W4. Each color in the distribution represents a specific layer index in the models, with blue indicating shallow layers closer to the data layer, and red representing deeper layers.

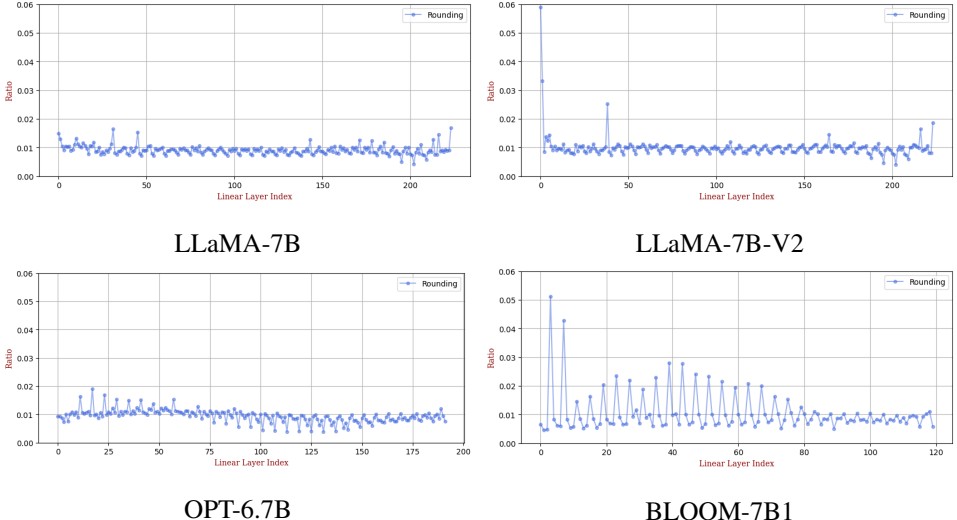

LLaMA-7B                    LLaMA-7B-V2

OPT-6.7B                    BLOOM-7B1

Figure 4: The correction between SignRound and salient activation channels

## F    RUNTIME

Table 13 provides a runtime comparison between GPTQ and our method. All measurements were conducted on a single NVIDIA A100 card with 80GB of memory. Although our method demonstrates slightly slower performance compared to GPTQ for models $\geq$ 30b, it remains well within acceptable limits for real-world deployment.

Table 13: Runtime in seconds at W4 for small models. Oursv2 denotes incorprating minmax tuning detailed in AppendixA. All the data of ours are tested with sequence length of 512. We haven't tested ours-addam for large models and there is no act-order option for bloom models.

| Type | LLaMA | | | | OPT | | | | BLOOM | |
|---|---|---|---|---|---|---|---|---|---|---|
| | 7B | 13B | 30B | 65B | 6.7B | 13B | 30B | 66B | 3B | 7B1 |
| GPTQ | 712 | 1240 | 3038 | 5635 | 841 | 1523 | 3014 | 6569 | 345 | 661 |
| Ours-adam | 937 | 1592 | - | - | 794 | 1479 | - | - | 431 | 837 |
| Ours | 730 | 1294 | 3902 | 6630 | 664 | 1184 | 3267 | 6385 | 362 | 695 |
| GPTQ-R | 929 | 1462 | 3062 | 5556 | 880 | 1513 | 3119 | 6374 | - | - |
| Oursv2 | 889 | 1605 | 4483 | 8183 | 825 | 1515 | 3882 | 7882 | 428 | 834 |

## G    DETAILED RESULTS OF SOME LLaMA MODELS

Detailed results of LLaMA7B, LLaMA13B, LLAMA7B-V2, and LLAMA13B-V2 can be found in Table 14, Table 15, Table 16, and Table 17 respectively.

Table 14: Accuracies(↑) of HellaSwag, WinoGrand, PIQA, LAMBADA and PPL(↓) of WikiText, PTB, C4 for **LLaMA-7B**, "Ours-2048" indicates that we have modified the sequence length of the calibration dataset from 512 to 2048.

|  |  | Hella. | Wino. | PIQA | Lamb. | Avg. | Wiki. | PTB | C4 |
|---|---|---|---|---|---|---|---|---|---|
|  | FP16 | 56.42 | 66.85 | 78.35 | 73.57 | 68.80 | 5.68 | 10.12 | 7.34 |
| W4G-1 | RTN | **54.96** | 67.25 | 77.31 | 70.00 | 67.38 | 6.29 | 11.25 | 8.12 |
|  | GPTQ | 52.25 | 63.85 | 74.59 | 68.12 | 64.70 | 6.59 | 12.02 | 8.64 |
|  | Ours | 55.28 | 66.14 | 77.64 | **73.14** | 68.05 | 6.12 | 10.88 | 7.84 |
|  | Ours-2048 | **54.96** | **67.56** | **77.80** | 72.13 | **68.11** | **6.05** | **10.87** | **7.76** |
| W4G128 | RTN | 55.86 | 65.75 | 77.58 | **72.25** | 67.86 | 5.96 | 10.54 | 7.70 |
|  | GPTQ | 54.09 | 64.09 | 77.26 | 69.86 | 66.33 | 6.29 | 11.11 | 8.02 |
|  | Ours | 55.92 | 66.30 | 78.07 | 72.07 | 68.09 | **5.86** | **10.49** | **7.56** |
|  | Ours-2048 | **55.98** | **66.77** | **78.29** | 71.78 | **68.21** | 5.88 | 10.52 | 7.58 |
| W3G128 | RTN | 53.17 | 63.14 | 75.73 | 67.71 | 64.94 | 7.01 | 12.83 | 9.18 |
|  | GPTQ | 47.10 | 59.91 | 72.58 | 53.58 | 58.29 | 8.28 | 16.84 | 10.45 |
|  | Ours | **53.98** | **66.06** | **76.61** | 69.82 | 66.62 | 6.93 | 11.67 | 8.30 |
|  | Ours-2048 | 53.45 | 65.67 | 76.55 | **71.08** | **66.69** | **6.52** | **11.60** | **8.26** |

Table 15: Accuracies(↑) of HellaSwag, WinoGrand, PIQA, LAMBADA and PPL(↓) of WikiText, PTB, C4 for **LLaMA-13B**, "Ours-2048" indicates that we have modified the sequence length of the calibration dataset from 512 to 2048.

|  |  | Hella. | Wino. | PIQA | Lamb. | Avg. | Wiki. | PTB | C4 |
|---|---|---|---|---|---|---|---|---|---|
|  | FP16 | 59.13 | 70.32 | 78.94 | 76.17 | 71.14 | 5.09 | 9.08 | 6.80 |
| W4G-1 | RTN | 57.96 | 68.19 | 78.18 | 70.95 | 68.82 | 5.53 | 9.78 | 7.23 |
|  | GPTQ | 57.96 | **70.24** | 77.97 | 73.84 | 70.00 | 5.33 | 9.48 | 7.13 |
|  | Ours | 58.02 | 69.61 | **78.94** | **75.74** | **70.58** | **5.32** | **9.37** | **7.05** |
|  | Ours-2048 | **58.13** | 69.69 | 78.67 | 74.95 | 70.36 | 5.34 | 9.49 | **7.05** |
| W4G128 | RTN | 58.43 | 70.32 | 79.33 | 75.32 | 70.85 | 5.26 | 9.29 | 6.94 |
|  | GPTQ | **58.79** | 70.56 | 79.33 | 75.00 | 70.92 | 5.21 | 9.28 | 6.92 |
|  | Ours | 58.62 | **71.35** | **79.76** | 75.98 | **71.43** | **5.19** | **9.18** | **6.90** |
|  | Ours-2048 | 58.47 | 70.56 | 79.22 | **76.23** | 71.12 | **5.19** | 9.19 | **6.90** |
| W3G128 | RTN | 56.39 | 67.56 | 77.20 | 69.63 | 67.70 | 5.88 | 10.58 | 7.86 |
|  | GPTQ | 56.58 | 67.96 | 78.07 | 72.31 | 68.73 | 5.64 | 9.95 | 7.54 |
|  | Ours | **57.04** | **69.14** | 77.86 | 74.33 | **69.59** | **5.53** | 9.81 | 7.39 |
|  | Ours-2048 | 56.62 | 68.82 | **78.13** | 74.42 | 69.50 | 5.57 | **9.76** | **7.37** |

Table 16: Accuracies(↑) of HellaSwag, WinoGrand, PIQA, LAMBADA and PPL(↓) of WikiText, PTB, C4 for **LLaMA-7B-V2**, "Ours-2048" indicates that we have modified the sequence length of the calibration dataset from 512 to 2048.

|  |  | Hella. | Wino. | PIQA | Lamb. | Avg. | Wiki. | PTB | C4 |
|---|---|---|---|---|---|---|---|---|---|
|  | FP16 | 56.69 | 67.17 | 78.35 | 73.88 | 69.02 | 5.47 | 32.91 | 7.26 |
| W4G-1 | RTN | 55.51 | 66.77 | 77.58 | 68.08 | 66.98 | 6.12 | **61.61** | 8.16 |
|  | GPTQ | 54.74 | 66.93 | 76.17 | 69.73 | 66.89 | **6.09** | NAN | **7.90** |
|  | Ours | 55.53 | 67.09 | 77.53 | **70.81** | 67.74 | 298.4 | 2677 | 11.20 |
|  | Ours-2048 | **55.63** | **67.96** | **77.64** | 69.92 | **67.79** | 196.7 | 2622 | 9.97 |
| W4G128 | RTN | **56.55** | 66.93 | 77.37 | 72.44 | 68.32 | **5.72** | **50.25** | 7.58 |
|  | GPTQ | 56.16 | **68.03** | **78.56** | 72.83 | **68.90** | 5.73 | NAN | **7.53** |
|  | Ours | 56.21 | 67.56 | 77.64 | 73.20 | 68.65 | 60.03 | 1786 | 8.16 |
|  | Ours-2048 | 55.97 | 67.09 | 77.15 | **73.57** | 68.45 | 48.91 | 1872 | 8.05 |
| W3G128 | RTN | **54.65** | 67.17 | 75.90 | 65.98 | 65.92 | 6.66 | **44.89** | 8.98 |
|  | GPTQ | 52.93 | 65.19 | 76.44 | 67.49 | 65.51 | **6.57** | NAN | **8.61** |
|  | Ours | 53.65 | 66.14 | **77.09** | 70.64 | 66.88 | NAN | 1159 | 9.88 |
|  | Ours-2048 | 53.91 | **67.32** | 76.33 | 71.12 | **67.17** | NAN | 1739 | 10.11 |

Table 17: Accuracies($\uparrow$) of HellaSwag, WinoGrand, PIQA, LAMBADA and PPL($\downarrow$) of WikiText, PTB, C4 for **LLaMA-13B-V2**, "Ours-2048" indicates that we have modified the sequence length of the calibration dataset from 512 to 2048.

|  |  | Hella. | Wino. | PIQA | Lamb. | Avg. | Wiki. | PTB | C4 |
|---|---|---|---|---|---|---|---|---|---|
|  | FP16 | 59.71 | 69.61 | 78.78 | 76.71 | 71.20 | 4.88 | 48.82 | 6.73 |
| W4G-1 | RTN | 58.56 | 69.30 | **78.45** | 74.36 | 70.17 | 5.20 | 58.57 | 7.14 |
|  | GPTQ | 57.81 | 67.48 | 77.86 | 73.84 | 69.25 | **5.16** | **52.46** | **6.87** |
|  | Ours | 58.63 | **69.61** | 77.91 | 73.98 | 70.03 | 9.15 | 66.80 | 7.72 |
|  | Ours-2048 | **58.87** | 68.67 | 78.07 | **75.90** | **70.38** | 6.51 | 60.35 | 7.28 |
| W4G128 | RTN | 59.12 | 69.46 | 78.02 | 76.29 | 70.72 | **4.98** | 52.22 | 6.87 |
|  | GPTQ | 59.22 | 68.51 | **78.84** | 76.13 | 70.68 | 4.99 | **51.59** | 6.87 |
|  | Ours | 59.20 | 69.14 | 78.35 | 76.56 | 70.81 | 5.80 | 51.92 | **6.84** |
|  | Ours-2048 | **59.25** | **70.48** | 78.29 | **76.81** | **71.21** | 5.00 | 51.78 | **6.84** |
| W3G128 | RTN | 57.03 | 67.56 | 77.86 | 72.37 | 68.70 | 5.52 | 62.33 | 7.58 |
|  | GPTQ | 56.99 | 66.69 | **78.40** | 72.85 | 68.73 | 5.45 | **55.09** | 7.54 |
|  | Ours | **57.29** | 68.90 | 77.37 | 75.22 | 69.70 | **5.35** | 59.57 | **7.35** |
|  | Ours-2048 | 57.20 | **70.88** | 78.13 | **75.35** | **70.39** | 10.38 | 66.22 | 7.92 |

