# OpenReview forum: "Optimize Weight Rounding via Signed Gradient Descent for the Quantization of LLMs"
_ICLR.cc/2024/Conference — Submitted to ICLR 2024_

### Official Review · Reviewer_TRJs · 2023-10-29

**Soundness:** 3 good
**Presentation:** 3 good
**Contribution:** 2 fair
**Rating:** 5
**Confidence:** 5

**Summary:**

This paper introduces SignRound, a lightweight and effective approach for optimizing weight rounding in Large Language Models (LLMs) with 3 and 4-bit weight-only quantization. SignRound leverages signed gradient descent and achieves remarkable results in just 400 steps, competing favourably with recent methods. Experiments on several datasets demonstrate the effectiveness of the proposed method.

**Strengths:**

The paper is well written and easy to follow. SignRound achieves significant results within only 400 steps, showcasing its efficiency.

**Weaknesses:**

The paper's novelty is limited, largely revisiting concepts like learning weight rounding previously introduced by AdaRound. It overlooks in-depth comparisons with established methods like AdaRound and FlexRound. Please see questions for details.

**Questions:**

1.	The paper's novelty is somewhat circumscribed, given its central focus on learning weight rounding—a concept previously introduced by AdaRound (Nagel et al., 2020). While SignRound's utilization of the signed gradient to refine the rounding function distinguishes it, the motivation behind prioritizing only the gradient direction (ignoring gradient magnitudes) remains ambiguous.

2.	The authors appear to have not fully accounted for certain pivotal baselines in their study. Specifically, AdaRound (Nagel et al., 2020) and FlexRound (Lee et al., 2023) stand out as established methods that delve into the realm of learning weight rounding. It would enhance the paper's comprehensiveness and comparative analysis if these methodologies were discussed and compared with SignRound.

3.	The performance comparisons between GPTQ and the proposed method are unfair since act-order was not enabled. This omission potentially skews the results, leading to instances where GPTQ underperforms compared to rounding-to-nearest (RTN) in some cases (W4 in Table 1).

4.	GPTQ, by utilizing second-order information, offers an efficient solution to the weight quantization problem. It would benefit for the authors to provide a clearer distinction of the advantages offered by the proposed method, especially when there are instances where it lags behind GPTQ, as evidenced in Tables 3, 4, and 5. Additionally, omitting a comparative analysis on training time introduces ambiguity, making it challenging to ascertain the relative efficiency of the two methods.

5.	Referring to Table 4, it is noteworthy that the proposed method underperforms compared to RTN for both W4G128 LLaMA-7B and W3G128 LLaMA-7B configurations. A more in-depth exploration or justification for this discrepancy would enhance the paper's clarity.

6.	The comparisons presented in Table 8 between AWQ and the proposed method appear unfair due to disparities in the calibration datasets. For a comprehensive assessment of the proposed method's efficacy, it would be better for the authors to provide fair comparisons.

---

> ### Author Response · Authors · 2023-11-16
> **Author Response**
>
> Thank you for your thorough comments! We will now discuss all of your questions.
>
> **1 Concerns about novelty**
>
> We have proposed a simple yet effective solution for the problem at hand by leveraging existing tools. This suggests that our approach, which may have been overlooked by the community, offers a novel solution. Furthermore, considering that the mainstream methods for weight-only quantization of LLMs are still GPTQ and equivalent transformations like AWQ, we are confident that our solution will prove valuable to the community and could serve as inspiration for further advancements.
>
> **2 Question about motivation**
>
> We have added the ablation study of optimizers in Section 4.3 and Table 6.
> The motivation is: The boundary of V in Eq.2 is constrained, within the range of [-0.5, 0.5], where only the value of altering the rounding is significant. These characteristics offer three advantages for signed gradient descent.  Firstly, the optimal value is not a single float but typically a large region, negating the need for the gradient magnitude to converge to an exact point. Secondly, due to the limited boundary, we can traverse this space within a constrained number of steps, while the gradient magnitude may vary significantly, making it challenging to determine an appropriate step size within a limited number of iterations. Thirdly, signed SGD is inherently intuitive, allowing for easy adjustment of the step size (learning rate). For instance, we employed the same optimizer hyperparameters in all of our experiments, which include 400 steps and a learning rate of 0.0025, with linear weight decay. This ensures that 400*0.0025/2=0.5 covers the range of [-0.5, 0.5]. Furthermore, the signed gradient descent method is quite lightweight, which is crucial since LLMs demand significant resources.
>
>  **3 Concerns of pivotal baselines**
>
>  Your suggestion is quite reasonable. However, as FlexRound is not open-sourced and our computation resource is limited, we are willing to quote their data directly if the comparison is somehow fair. For FlexRound, the most relevant to ours is their Table 24. They report wikitext2 perplexity based on training it on wikitext2 dasetset with 5000 steps, and for ptb, they also train on ptb dataset with 5000 steps. This approach is prone to overfitting and may not be quite useful for pretrained LLMs. Furthermore, they tuned hyperparameters for each model. Based on these factors, we believe the comparison is not fair, and as a result, we decided not to compare their data in the paper.
>
> **4 Concerns of GPTQ act-order**
>
> Thank you for the information, and we apologize for not incorporating the update introduced in June 2023. We had cloned the code some time back, and misunderstood the no grouping setting. We have re-run the experiments at W4G-1 and have included the new data in Tables 1, 3, 8, and 9. The results consistently highlight the advantages of SignRound. We plan to include additional data in the future.
>
> **5 Questions about distinction of the advantages**
>
> Although GPTQ is well motivated by optimal brain surgery, it is our understanding that it inherits two disadvantages. For example, let's split the tensor of weight into A and B.  After quantizing A, GPTQ adjusts B to minimize the error introduced by the quantization of A. However, if quantizing A results in a significant error, such as when quantizing A to 1 bit, it becomes challenging to adjust B. In contrast, SignRound keeps reducing the error of quantized A in mind. Additionally, after quantizing B, there are no remaining weights available to adjust and compensate for this error, whereas SignRound does not face this issue.
>
> Given the numerous models and scenarios, we believe that SignRound could be beneficial, and we think in many scenarios, SignRound could be combined with other SOTA methods.
>
>
>
> **6 Questions about running time**
>
> We have included additional data in Table 13. Since our code has been optimized,  we use the latest data. For instance, when quantizing llama-65b, SignRound  takes 1.85 hours.
>
> **7 Questions about scenarios inferior to RTN**
>
> To be honest, we were unable to determine the root cause of the issue. We have two hypotheses that may explain the problem. Firstly, the output reconstruction task is  be a proxy task, and lower error may not always provide an advantage for the real task. Secondly, the limited size of the calibration dataset might introduce some bias.
>
> **8 Concerns about fair comparison with AWQ**
>
> we have update the data of AWQ calibrated with pile-10k in Appendix B

---

> > ### Comment · Reviewer_TRJs · 2023-11-22
> > **Response to Rebuttal**
> >
> > Thank you for your response. However, I still have several concerns:
> >
> > **Clarity of Novelty**: The novelty of the proposed method remains unclear in the current manuscript. It is crucial to address this issue by conducting a comprehensive performance comparison between the proposed method and pivotal baselines, such as AdaRound (Nagel et al., 2020) and FlexRound (Lee et al., 2023). Demonstrating the advantages and superiority of your approach over these well-established techniques is essential to establish its uniqueness and effectiveness.
> >
> > **Fairness in Comparison with GPTQ**: The comparison between the proposed method and GPTQ raises concerns about fairness. GPTQ relies on a calibration dataset of 128 samples, while the proposed method employs a significantly larger dataset of 512 samples.
> >
> > **Inferior Results Compared to RTN**: The observed inferior results of the proposed method compared to RTN are a significant concern.

---

> > > ### Author Response · Authors · 2023-11-22
> > > **Author Response**
> > >
> > > Thank you for your feedback.
> > >
> > > 1 Your concern is quite reasonable.  As mentioned earlier, we haven't compared with them due to several issues. However, we have demonstrated that sign gradient descent can perform well in various scenarios without needing to tune any optimizer hyperparameters. Additionally, in many instances of W4G-1 and W4G128, we have achieved results that are quite comparable to FP16's with the assistance of v2. We hypothesize that even if other methods are superior in some scenarios, their advantages in these scenarios are minimal and likely slower
> > >
> > > 2  We could add an ablation study for this after the rebuttal, however, we use the default 128 samples for a fair comparison. Various algorithms have different preferences for hyperparameters based on runtime considerations.   In GPTQ, the batch size is equivalent to the total samples, and scaling it to 512 samples would lead to a 1X increase in runtime. Signround also made compromises on runtime considerations, such as reducing the sequence length from 2048, as used in GPTQ, to 512 or 256. Tables 9, 17, 16, and 14 all demonstrate that a sequence length of 2048 holds advantages, and in some cases, significant advantages.  We also enabled AMP.
> > >
> > >
> > > 3  Given our earlier explanation and the fact that GPTQ also experiences the same issue, we believe this phenomenon is likely to occur with other algorithms, particularly when reducing the group size. For instance, the opt125M at W4G18, RTN could achieve better results than FP16, while the most robust algorithm using output reconstruction will be equivalent to FP16.

---

### Official Review · Reviewer_yH6X · 2023-10-30

**Soundness:** 2 fair
**Presentation:** 2 fair
**Contribution:** 1 poor
**Rating:** 5
**Confidence:** 4

**Summary:**

For better weight-only LLM quantization, this work proposes to optimize the rounding of weights with the block-wise reconstruction error as the objective. Following previous smart rounding work such as AdaRound, it optimizes continuous variables that will be added onto the scaled weights before rounding. This work also emphasizes the need to use "signed gradient descent" in the rounding variable optimization, which only exploits the gradient direction instead of the magnitude. This work conducted experiments on quantizing LLaMA v1, v2, BLOOM, and OPT models to W3 and W4 with different group sizes.

**Strengths:**

* Motivation: How to make weight-only LLM quantization work for fewer bits is worth studying, especially for edge scenarios with very limited memory.
* Reasonable pathway: Applying smart-rounding PTQ methods is a reasonable pathway.
* The experiments are conducted with different model families.
* The paper is easy to understand.

**Weaknesses:**

* Unclear logic of applying signed gradient descent: The paper said "prefer the signed gradient descent method to tackle the issue of sub-optimal rounding" without intuitive logic description, theoretical justification, or experimental verification of this technique.
* Marginal improvements: The method shows improvements on relatively smaller models, but on larger models (>7B), it achieves marginal improvements over GPTQ or RTN.

**Questions:**

See the weakness part

---

> ### Author Response · Authors · 2023-11-16
> **Author Response**
>
> Thank you very much for your comments! We will now discuss all of your questions.
>
> **1 Concerns about unclear logic of applying signed gradient descent**
>
> We have added the ablation study of optimizers in Section 4.3 and Table 6.
> The motivation is: The boundary of V in Eq.2 is constrained, within the range of [-0.5, 0.5], where only the value of altering the rounding is significant. These characteristics offer three advantages for signed gradient descent.  Firstly, the optimal value is not a single float but typically a large region, negating the need for the gradient magnitude to converge to an exact point. Secondly, due to the limited boundary, we can traverse this space within a constrained number of steps, while the gradient magnitude may vary significantly, making it challenging to determine an appropriate step size within a limited number of iterations. Thirdly, signed SGD is inherently intuitive, allowing for easy adjustment of the step size (learning rate). For instance, we employed the same optimizer hyperparameters in all of our experiments, which include 400 steps and a learning rate of 0.0025, with linear weight decay. This ensures that 400*0.0025/2=0.5 covers the range of [-0.5, 0.5]. Furthermore, the signed gradient descent method is quite lightweight, which is crucial since LLMs demand significant resources.
>
> **2 Concerns about marginal improvements over 7B models in comparison to RTN and GPTQ**
>
> We are grateful for the acknowledgment of the improvement in smaller models. However, it's important to note that for larger models, the improvement may seem marginal if we overlook the small gap between RTN/GPTQ with FP16, typically $<$ 2 points in most cases. We present the average improvements of all the quantization configurations introduced by SignRound (without minmax tuning) over RTN, GPTQ (choosing the best with actorder and without actorder if data is available), and FP16, respectively. When considering the gap to FP16, we observe that SignRound achieves notable improvement over RTN. In comparison to GPTQ, we demonstrate advantages on 13B models. For models larger than or equal to 30B, SignRound is comparable with GTPQ. All the data is presented in absolute values.
>
> Advancements of SignRound over RTN, GPTQ and FP16 as tuples of  (ours-RTN, ours-GPTQ, ours-FP16):
>
> Llama13b (1.41, 0.63, -0.61),
>
> Llama13bv2 (0.32,0.29, -1.02),
>
> LLam13bv2-ours-seq2048 (0.80, 0.77,-0.54),
>
> OPT13B(5.28,-0.21, -0.71),
>
> Llama30b(0.32, 0.07,-0.89),
>
> Llama60b(0.01,-0.40,-1.67),
>
> opt30b-ours-seq512(2.67, -0.10, -0.82),
>
> opt66b(30.84, 0.15, -1.14).
>
> Please kindly note that SignRoundv2(with minmax tuning) could achieve better result in most scenarios as shown in Table 1 and Table 9, especially for 13B models.
>
> Furthermore, since our method is based on lightweight training and our default hyperparameters are identified through experiments on smaller models, there is potential for further tuning.

---

> > ### Comment · Reviewer_yH6X · 2023-11-23
> > **Thanks for the response and some follow-up**
> >
> > Thanks for the authors' response and revised paper (a suggestion is to make all the revisions in blue, it's not clear now). I  decided to keep my original ratings due to the following reasons.
> >
> > (1) For the response to the first question:
> > > Due to the limited boundary, we can traverse this space within a constrained number of steps, while the gradient magnitude may vary significantly, making it challenging to determine an appropriate step size within a limited number of iterations.
> >
> > The logic is not so clear to me. Do you mean when the number of iterations is small, Signed-SGD is better than other optimizer?
> >
> > > Signed SGD is inherently intuitive, allowing for easy adjustment of the step size (learning rate)
> >
> > I'm not sure what does ``inherently intuitive'' mean. And isn't this point the same point as the second one?
> >
> > From the newly added Table 6, the improvements brought by Signed SGD seem not consistent nor significant.
> >
> > (2) For my second concern. I appreciate the fact that the authors report all the numbers, no matter whether their method is better or worse than GPT-Q. Also, I note the authors developed an extension of their method "ours-v2" during the rebuttal period. But still, it seems the proposed method doesn't achieve a significant and consistent improvement over GPT-Q. This makes me wonder "will we actually use this method instead of GPT-Q".
> >
> > Another minor thing is that I noted some minor writing issues, e.g., wrong quotation marks, inconsistent abbreviations in the result table. The authors can fix that in the future.

---

> ### Author Response · Authors · 2023-11-23
> **Thank you**
>
> Thank you for taking the time to review, and we respect your decision. We want to clarify a little.
>
> 1 We apologize for any confusion. In the case of signed SGD, at each step, we precisely know the step size (-lr or lr), and each parameter has an equal chance to traverse the space. However, adaptive algorithms rely on gradient magnitude to determine the step size, resulting in some parameters converging quickly while others converge slowly. This may lead to certain parameters having little chance to converge within a limited number of steps. Therefore, choosing default hyperparameters becomes challenging. In works where adaptive optimizers are applied , lr and iterations need to be tuned if numerous scenarios have been validated.
>
>  We have demonstrated that sign gradient descent can perform well in many different scenarios without needing to tune any optimizer hyperparameters. Additionally, in many instances of W4G-1 and W4G128, we have achieved results that are quite comparable to FP16's with the assistance of v2. We hypothesize that even if other optimizers are better in some scenarios, their advantages in these specific scenarios are minimal. Due to the little resource needed, sign gradient descent holds advantages.
>
>
>
> 2 With numerous models and metrics for LLMs, it is challenging for one algorithm to  outperform the others in all scenarios. However, we have demonstrated that signroundv2 outperforms GPTQ in 34 out of 39 scenarios for small models and 6 out of 8 scenarios for large models, with our best result achieved in zero-shot tasks.
>
>
>
> 3 Thank you very much for the information. We will check the paper several times.

---

### Official Review · Reviewer_GzCp · 2023-10-31

**Soundness:** 4 excellent
**Presentation:** 4 excellent
**Contribution:** 2 fair
**Rating:** 6
**Confidence:** 5

**Summary:**

This work develops a signed gradient decent for the quantization of large language model weights. It compares against other approaches that are more complicated algorithmically and achieve slightly better results.

Recommendation: This is a solid contribution that I would rather see accepted than rejected.

**Strengths:**

- simple approach will make it easier to develop more complicated methods. This is a large advantage over GPTQ which is a good foundation for future methods.
- evaluation is quite extensive, leaving little doubt that the method works well
- shows that the Hessian approach from GPTQ does not add too much unique value, but mostly reduces the samples needed for good performance. This is a very valuable insight that will quite future quantization work.

**Weaknesses:**

- while the simplicity is an advantage of this method, it can also be seen as a disadvantage. However, I would like to highlight for the AC and reviewers that the main goal of the paper is to simplify a complicated algorithm (GPTQ), and the authors succeed
- not competitive with other more extensive methods. However, other methods cannot be used as a base optimization method for finding quantization. As such, this approach is more useful for future work

**Questions:**

Comments:
 - before equation 6, the equation is missing an "s"

Questions:
- why is there such high C4 perplexity for transformer block quantization for W3G128? Do the numbers indicate some form of instability during sign SGD?
- Why is the runtime so slow? Are you building on the GPTQ codebase?

---

> ### Author Response · Authors · 2023-11-16
> **Author Response**
>
> Thank you for your kind review. e will now discuss all of your questions.
>
> **1 Question about poor c4 perplexity**
>
> With the observation that the c4 perplexity of GPTQ for Llamav2 is reasonable, while its ptb perplexity is shown as NAN in Table 16, and SignRound has the same issue,  we hypothesize this is likely because perplexity is highly sensitive to outliers. The equation is represented as $exp(-(sum(log p))/seqlen)$, where a low probability p for a single token results in a high perplexity value.
>
> It appears that SignRound introduces a higher number of outliers, and at present, we haven't been able to pinpoint the underlying cause. However, this issue could be  mitigated by incorporating minmax tuning as shown in Appendix and Table 3. Alternatively, tuning the hyperparameters, which we have confirmed to be effective in some outlier scenarios, could also alleviate this issue.
>
> **2 Question about slow runtime**
>
> The reported runtime reflects the cost of generating the quantization model, and our code is not built on the GPTQ codebase.  We have provided the code for your review.  Additionally, we have recently made some optimizations to the code, resulting in a reduction of the llama7b runtime from 899s to 730s.  It's worth noting that we are not experts in GPU performance, and there may still be  room for further enhancement.
>
> **3 Concerns about simplicity**
>
> In Appendix A and Table 1, we demonstrate the seamless incorporation of weight minmax tuning using sign gradient descent, showcasing notable improvements with only a 0.2X runtime overhead. Our method's simplicity is a key advantage, and we believe there is still a room for further improvement.
>
> **4 Concerns about more baselines**
>
> We sincerely appreciate your understanding. We have updated some data in comparison to AWQ in Appendix B.1.
>
> **5 Comment of typo**
>
> Apologies for any confusion, but to our understanding, equation 6 should not include 's', while equation 7 should incorporate 's'.

---

> > ### Comment · Reviewer_GzCp · 2023-11-23
> > **Thank you.**
> >
> > Thank you for your candid rebuttal. I get the impression that you did careful work here and I think your work is a great contribution. It has some flaws though, and if I could give you a score of 7, I would raise my score to that. However, I do not think with the flaws this work is quite a score of 8. As such, I will keep my current score.

---

> > > ### Author Response · Authors · 2023-11-23
> > > **Thank you**
> > >
> > > Thank you for your kind acknowledgment and for taking the time to review; we truly appreciate it.

---

### Official Review · Reviewer_vB5y · 2023-10-31

**Soundness:** 2 fair
**Presentation:** 2 fair
**Contribution:** 2 fair
**Rating:** 5
**Confidence:** 4

**Summary:**

* The paper proposes to optimize the layer-wise rounding problem that occurs for LLM PTQ using signed gradient descent.
* The method is evaluated across various models and tasks.

**Strengths:**

* The paper is easy to follow.
* The paper conducts a large number of experiments across various models, tasks and quantization setting. Further, it also consider state-of-the-art LLMs like Llama2 in addition to older ones like OPT and BLOOM.
* SignRound appears to bring some performance improvements relative to GPTQ, in particular on smaller models and for zero-shot tasks.
* I also like that the paper includes also handful of unfavorable results to provide a more complete.

**Weaknesses:**

* The paper essentially seems to apply signed gradient descent (which is not new) to the standard layer-/block-wise rounding problem considered by various LLM PTQ papers. Hence, the overall novelty is low.
* GPTQ Activation-reordering can also be performed without any impact on inference performance (see the official GPTQ repo, option `--static-groups`). Further, if there is no grouping, reordering has no impact on inference. LLaMa1-7B and OPT-66B are known to be GPTQ outliers, for which reordering should be enabled to conduct a fair comparison.
* The paper argues that signed gradient descent is preferable over standard straight-through QAT (applied to the layer-wise quantization problem, like ZeroQuant) for this application, but does not provide any ablation studies supporting that point.
* Based on Table 5, it appears that for the largest and most interesting models for compression applications, SignRound seems to perform very similar to GPTQ and in some cases even worse.
* The code is not available in the Supplementary material.

Overall, I am currently leaning towards rejection as the novelty is rather low and the experimental results not quite strong enough (and with some problems/questions) to make up for it.

**Questions:**

* Is GPTQ using the same amount of samples as SignRound in your comparisons?
* Do you have any explanation for the surprisingly poor perplexity performance on Llama" models in Table 3? Related to that, how comes that those models still exhibit comparable or better ZeroShot performance.
* The runtime comparisons in Table 12 stop at 13B size, how long does your method take for the largest models?

---

> ### Author Response · Authors · 2023-11-16
> **Author Response**
>
> Thank you for your detailed comments! We will now discuss all your questions.
>
> **1 Question about GPTQ samples**
>
> We use the default 128 samples for a fair comparison. In GPTQ, the batch size is equivalent to the total samples, and scaling it to 512 samples would lead to a 1X increase in runtime. In contrast, our runtime is comparable to their default setting, as demonstrated in table 13 of Appendix F.
>
> **2 Question about poor c4 perplexity**
>
> With the observation that the c4 perplexity of GPTQ for llamav2 is reasonable, while its ptb perplexity is shown as NAN in Table 16, and SignRound has the same issue,  we hypothesize this is likely because perplexity is highly sensitive to outliers. The equation is represented as $exp(-(sum(log p))/seqlen)$, where a low probability p for a single token results in a high perplexity value. On the other hand, the accuracy does not exhibit such issues, as each value falls within the range of [0,1].
>
> It appears that SignRound introduces a higher number of outliers, and at present, we haven't been able to pinpoint the underlying cause. However, this issue could be  mitigated by incorporating minmax tuning as shown in Appendix A and Table 3. Alternatively, tuning the hyperparameters, which we have confirmed to be effective in some outlier scenarios, could also alleviate this issue.
>
> **3 Question about runtime**
>
> We have added supplementary data to Table 13. Our code has been optimized, allowing us to utilize the latest data. For example, when quantizing llama-65b, SignRound only takes 1.85 hours.
>
> **4 Concerns about novelty**
>
> We have proposed a straightforward yet effective solution for the problem at hand, utilizing existing tools. This suggests that our approach, which may have been overlooked by the community, offers a novel solution. Furthermore, considering that GPTQ and equivalent transformations like AWQ are still the mainstream methods for weight-only quantization of LLMs, we are confident that our solution will be valuable to the community and could inspire further advancements.
>
> **5 Concerns about GPTQ act-order**
>
> Thank you for the information, and we apologize for not incorporating the update introduced in June 2023. We had cloned the code some time back, and misunderstood the no grouping setting. We have re-run the experiments at W4G-1 and have included the new data in Tables 1, 3, 8, and 9. The results consistently highlight the advantages of SignRound. We plan to include additional data in the future.
>
> **6 Concerns about ablation study of optimizers and the motivation**
>
> We have added the ablation study in Section 4.3 and Table 6.
> The motivation is: The boundary of V in Eq.2 is constrained, within the range of [-0.5, 0.5], where only the value of altering the rounding is significant. These characteristics offer three advantages for signed gradient descent.  Firstly, the optimal value is not a single float but typically a large region, negating the need for the gradient magnitude to converge to an exact point. Secondly, due to the limited boundary, we can traverse this space within a constrained number of steps, while the gradient magnitude may vary significantly, making it challenging to determine an appropriate step size within a limited number of iterations. Thirdly, signed gradient descent is inherently intuitive, allowing for easy adjustment of the step size (learning rate). For instance, we employed the same optimizer hyperparameters in all of our experiments, which include 400 steps and a learning rate of 0.0025, with linear weight decay. This ensures that 400*0.0025/2=0.5 covers the range of [-0.5, 0.5]. Furthermore, the signed gradient descent method is quite lightweight, which is crucial since LLMs demand significant resources.
>
> **7 Concerns about accuracy for largest models comparing to GPTQ**
>
> Larger models typically exhibit greater resilience to compression, and recent state-of-the-art methods often deliver results that closely match those of FP16, leaving little room for improvement.
>
> We present the average improvements of all the quantization configurations introduced by SignRound (without minmax tuning) over RTN, GPTQ (choosing the best without actorder and with actorder if we have the data), and FP16, respectively, as tuples of (ours-RTN, ours-GPTQ, ours-FP16):
>
> - Llama30b (0.32, 0.07, -0.89)
> - Llama60b (0.01, -0.40, -1.67)
> - opt30b-ours-seq512 (2.67, -0.10, -0.82)
> - opt66b (30.84, 0.15, -1.14).
>
> It is important to note that SignRoundv2 (with minmax tuning) could achieve better results in most scenarios, as demonstrated in Table 1 and Table 9.
>
> Furthermore, given that our method is based on lightweight training and our default hyperparameters are identified through experiments on smaller models, there is potential for further tuning.
>
> **8 Concerns about code**
>
> We have uploaded the code

---

> > ### Comment · Reviewer_vB5y · 2023-11-22
> > **Reviewer Response**
> >
> > Thank you for the clarifications!
> >
> > * I am still not convinced that signed gradient descent is better than standard QAT, e.g. as used by ZeroQuant, when both are fully tuned. I believe this would be key to establish very clearly, since the motivation of the paper hinges on this. Section 4.3 states AdamW performs comparable to SignRound but is 20-30% slower and requires more memory. Given that the overall runtime of this process is at most a few hours and optimization is performed layer-wise, neither of those strikes me as a particularly big advantage. In general, this Section still feels quite preliminary, leaving many questions open, and should be extended significantly.
> >
> > * In terms of accuracy comparisons, I think it would be import to ensure matching setups as much as possible, in order to identify precisely where improvements are coming from. Using 128 samples for GPTQ and 512 for SignRound is really not ideal in that regard; especially also since some of the improvements are quite small.
> >
> > * For large models, it also seems to me that in Tables 9 & 10, GPTQ/GPTQ-R wins in 13/16 cases (the v2 results are not comparable since grid-clipping could also be applied to GPTQ in some form); further confirming my initial concerns based on Table 5.
> >
> > For those reasons I maintain my initial score.

---

> ### Author Response · Authors · 2023-11-22
> **Author Response**
>
> Thank for your feedback and we respect your decision. However,  we still want to make some clarifications
>
> 1   Based on our understanding, QAT requires more resources and involves hyperparameter tuning, such as learning rate and iterations, as reported in zeroquant.  For the optimizer part, we have demonstrated that sign gradient descent can perform well in many different scenarios without needing to tune any optimizer hyperparameters. Additionally, in many instances of W4G-1 and W4G128, we have achieved results that are quite comparable to FP16's with the assistance of v2. We hypothesize that even if other optimizers are better in some scenarios, their advantages in these specific scenarios are minimal.
>
> 2 Various algorithms have different preferences for hyperparameters based on runtime considerations. Signround also made compromises, such as reducing the sequence length from 2048, as used in GPTQ, to 512 or 256, enabling AMP, and controlling the iteration to 400. Tables 9, 17, 16, and 14 all demonstrate that a sequence length of 2048 holds advantages, and in some cases, significant advantages."
>
> 3  We are quite close to the GPTQ in some scenarios, especially for accuracies. In our understanding, adding minmax tuing to GPTQ, regardless of how much benefit it can bring to GPTQ, at least compared to signround, it will significantly increase its runtime.

---

### Author Response · Authors · 2023-11-16
**Paper Revision Summary**

Dear reviewers,

We extend our gratitude for your valuable feedback and want to highlight that we have recently submitted a revised version of the paper to address your concerns that you raised. The latest revision includes the following updates:

1. Include an optimizer ablation study in Section 4.3 and provide a detailed explanation of its motivation at the end of the Introduction, before the contribution part.

2. We introduce min-max tuning through sign gradient descent and provide a detailed explanation in Appendix A. The corresponding data is included in Tables 1, 2, 9, and others, showcasing only ~20\% runtime overhead which is typically affordable.

3. We have added GPTQ actorder data for all the zero shot tasks we tested, In summary, when compared to
best(GPTQ,GPTQ-actorder) for models <=13B, our approach surpasses it in 27 out of 39 scenarios. Additionally, when incorporating signroundv2, we outperform it in 34 out of 39 scenarios.  For models>=30b, signroundv2 could be comparable or better than best(GPTQ, GPTQ-actorder)

4. Supplementary runtime data has been included in Table 13 of Appendix F.

5. Data of AWQ calibrated with Pile-10k has been updated in Appendix B.1.

6. Code has been uploaded.

We are eager to engage in further discussions regarding this paper revision and our responses in general.
Best regards, The authors

---

### Meta-Review · Area_Chair_3z7j · 2023-12-10

**Metareview:**

This paper proposes a sign gradient descent method to optimize the weight rounding for quantizing llms, which achieves superior performance than gptq and awq for several common llms including opt, llama-1, and llama-2. One notable merit of this paper is its simplicity. Weaknesses include limited novelty, marginal improvements, and lack of comparisons over previous works AdaRound and FlexRound. Overall, the reasons to reject slightly overweighs the reasons to accept.

**Justification For Why Not Higher Score:**

limited novelty, marginal improvements, and lack of comparisons over previous works AdaRound and FlexRound

**Justification For Why Not Lower Score:**

N/A

---

### Decision · Program_Chairs · 2024-01-16

Reject